



# Implementation and application of an improved phase spectrum determination scheme for Fourier Transform Spectrometry

Frank Hase[1], Paolo Castracane[5], Angelika Dehn[5], Omaira Elena García[7], David W. T. Griffith[3], Lukas Heizmann[6], Nicholas B. Jones[3], Tomi Karppinen[4], Rigel Kivi[4], Martine de Mazière[2], Justus Notholt[6], Mahesh Kumar Sha[2]

[1]IMKASF, Karlsruhe Institute of Technology (KIT), Eggenstein-Leopoldshafen, 76344, Germany

[2]Royal Belgian Institute for Space Aeronomy (BIRA-IASB), Ringlaan 3, 1180 Brussels, Belgium

[3]University of Wollongong, Wollongong, Australia

[4]Space and Earth Observation Centre, Finnish Meteorological Institute, Sodankylä, Finland

[5]European Space Agency, ESA/ESRIN, Frascati RM, Italy

[6]Institute of Environmental Physics, University of Bremen, Bremen, Germany

[7]Izaña Atmospheric Research Centre (IARC), State Meteorological Agency of Spain (AEMet), Santa Cruz de Tenerife, Spain

*Correspondence to*: Frank Hase (frank.hase@kit.edu)

**Abstract.** Correct determination of the phase spectrum is a highly relevant task in Fourier Transform Spectrometry for concluding which spectral distribution connects with the measured interferogram. We present implementation of an improved scheme for phase determination in the operational Collaborative Carbon Column Observing Network (COCCON) processor. We introduce a robust unwrapping scheme for retrieving a connected phase spectrum at intermediate spectral resolution, which uses all spectral positions carrying enough signal to allow a significant determination of the phase. In the second step, we perform a least squares fit of model parameters of a suited analytical phase spectrum model through all reliable phase values constructed in the first step. The model fit exploits the fact that we expect the phase to be spectrally smooth. Still, it can be refined to reflect specific characteristics inherent to the optical and electronic layout of the interferometer. The proposed approach avoids the problems of the classical phase reconstruction method, which enforce a spectrally smooth phase by directly limiting spectral resolution when calculating the complex phase. Thereby, the phase is created from a very low number of interferogram points around the centerburst of the interferogram, which results in a suboptimal noise propagation from the interferogram into the spectral domain. Moreover, the interpolation of the phase spectrum across spectral subsections with reduced spectral signal is not well behaved and results depend strongly on the numerical apodization function used for creating the low-resolution phase.



## 1 Introduction

Fourier Transform Spectrometry is an important technique for remote observation of atmospheric composition, especially in the near and mid infrared spectral regions (then mostly referred to as Fourier Transform Infrared or shortly FTIR spectroscopy). Ground-based networks contribute to the long-term monitoring of chemical composition, as the Network for the Detection of Atmospheric Composition Change (NDACC) network [De Mazière et al., 2018], and the Total Carbon Column Observing Network (TCCON) [Wunch et al., 2011] and the COllaborative Carbon Column Observing Network (COCCON) [Frey et al., 2019; Sha et al., 2020; Alberti et al., 2022], which focus on the provision of precise and accurate observations of column-averaged greenhouse and other climate and air quality relevant gas abundances. Moreover, highly successful space borne sensors as Michelson Interferometer for Passive Atmospheric Sounding (MIPAS) onboard the Environmental Satellite (ENVISAT) [Fischer et al., 2008], Atmospheric Chemistry Experiment – Fourier Transform Spectrometer (ACE-FTS) onboard SCISAT [Bernath and al., 2005], and the Thermal And Near infrared Sensor for carbon Observation – Fourier Transform Spectrometer (TANSO-FTS) onboard Greenhouse gases Observing SATellite (GOSAT) [Yokota et al., 2009] and its successors have proven the usefulness of  FTIR spectrometry for atmospheric observations. Recently, the airborne imaging FTIR sensor Gimballed Limb Observer for Radiance Imaging of the Atmosphere (GLORIA) for chemical and thermal limb imaging has been realized [Friedl-Vallon et al., 2014] and the imaging FTIR satellite mission Changing Atmosphere Infrared Tomography (CAIRT) derived from GLORIA is under phase A study by ESA [https://www.cairt.eu].

All FTIR spectrometers have in common that they use a two-beam interferometer for creating modulated intensity levels as a function of the path difference between the two arms of the interferometer. The path difference is varied as function of time, and during such a scan, the variable intensity is recorded by a detector element. By use of a co-recorded reference modulation generated from a reference laser fed through the same interferometer, the variable intensity level recorded by the infrared detector as function of time can be sampled as function of optical path difference $x$. It can be shown that the Fourier Transform of the AC-coupled interferogram is associated with the spectral distribution of the incident radiation. If the interferogram $I(x)$ would be symmetric around a common zero path difference (ZPD) of the interferometer for any wavenumber $\nu$, the spectral radiance as function of wavenumber $S(\nu)$ would be connected with the interferogram via a simple cosine transform:

$$S(\nu) \int_{x=-\infty}^{+\infty} I(x) cos(2\pi\nu x) dx \qquad (1)$$

We only claim a proportionality here for any selected wavenumber position, because from the practical viewpoint, the determination of radiances in absolute units requires proper calibration measurements using reference sources providing a known radiance level. This is a very laborious task and it is difficult to achieve sub-percent accuracy in the realization of absolute units. In case of emission spectroscopy, this task needs to be solved, while atmospheric absorption spectroscopy



generally omits this procedure. In the case of absorption spectroscopy, the quantitative trace gas analysis is built on the local
contrast between absorption lines and adjacent continuum (assuming that the spectrometer offers sufficient spectral
resolution for resolving individual lines). Then, by assuming that the spectrally variable sensitivity of the spectrometer,
created by optical, detector, and electronic characteristics is spectrally smooth, no attempt is made for achieving ordinate
calibration. A section of the measured spectrum used for the trace gas analysis is then treated as a transmission spectrum, and
an empirical fit of continuum background is included in the analysis scheme. We do not further follow the problem of
ordinate calibration here, because it is not related to our aim of an improved phase reconstruction, which, however, can be
used for both absorption and emission spectroscopy.

In equation (1), we have extended the integration over all optical path differences. In practice, only a limited section up to a
maximum optical path difference (MPD) is accessible. The truncation of the interferogram is equivalent to a multiplication
with a boxcar function. This spectral response inherent to an FTIR spectrometer can be adjusted by applying a numerical
weighting function along the interferogram (the process of apodization). A proper description of the instrumental line shape
(ILS) is further complicated due to the presence of practical imperfections of the interferometer. Finally, we do not further
follow the problem of spectral ordinate calibration here, because it, too, is not closely related to our aim of an improved
phase reconstruction.

In order to provide a proper idea of the practical method of FTIR spectroscopy here, we further need to mention that the data
recording and processing is digital. An analogue-to-digital (ADC) converter is used to generate a digitized signal from the
detector signal. While sample-and-hold ADCs triggered by the laser sampling were used in the past, many manufacturers of
FTIR spectrometers today use widely available audio ADCs which offer high digitization depth (e.g. 24 bit) and add a final
interpolation step from the raw sampling equidistant in time domain into a sampling record equidistant in space [Brault,
1996]. In any case, the signal to be processed is discretely sampled, and in practice fast computational schemes for doing
discrete Fourier transforms are applied. Due to the discrete sampling process, integrals as shown in equation (1) become
sums and the bandwidth of the recorded signal needs to be properly limited in order to avoid aliasing.

A final aspect, which is closely connected to the considerations developed hereinafter, is the origin of the phase spectrum.
Due to residual optical asymmetry of the beamsplitter and possibly between the arms of the interferometer and due to
frequency dependent electronic delays, the resulting interferogram tends to be asymmetric and a global ZPD position
common to all wavenumbers does not exist. The electronic delays introduce both a shift between the laser reference and the
signal, as well as frequency-dependent delays in the infrared signal. This requires treatment of the Fourier Transform of the
real-valued interferogram as a complex quantity (so arising out of cosine and sine contributions) and thereby gives birth to
the concept of the phase spectrum. In complex notation, we can state





$$s(v) = |s(v)|e^{i\varphi(v)} = \int_{x=-\infty}^{+\infty} I(x)e^{-i2\pi vx}dx \qquad (2)$$

The uncalibrated signal $s(v)$ now is a complex quantity. It can be separated into amplitude and phase $\varphi(v)$. The phase
spectrum $\varphi(v)$ describes how the phase angle of the harmonic oscillations which make up the interferogram evolves as
function of wavenumber. From the instrumental viewpoint, we expect the phase spectrum to be spectrally smooth, as the
impacting factors (optical dispersion and electronic delays) typically vary slowly as function of frequency.

The smoothness of the phase spectrum in near and mid-infrared FTIR spectroscopy is verified empirically on scales of
several to tens or even hundreds of wavenumbers (cm$^{-1}$). Given this, the simple approach of interpreting the absolute value of
the resulting complex spectrum as the measured spectral signal is clearly suboptimal in the presence of noise in the
interferogram. The assumption of uncorrelated white noise typically is adequate. This noise maps into white noise in the
complex spectrum. A contribution of $1/f$ noise might increase the noise amplitude towards low frequencies, and at very low
frequencies, source noise might become dominant. Therefore, working at higher scan speeds is generally preferred.

The assumption of a spectrally smooth phase allows to separate at each spectral position the complex spectrum into two
orthogonal components: the component along the direction we expect the spectral signal to be oriented, and the component
orthogonal to this direction. So, by exploiting the concept of a spectrally smooth phase, the noise mapped into the orthogonal
component can be avoided, only the noise along the signal component is unavoidable. Moreover, this approach avoids the
spectral noise floor of becoming a positive bias in opaque spectral subsections, as it would occur when simply using the
absolute value of the complex spectrum.

In order to make the scheme of a smooth phase a working concept, we not only rely on the assumption that it actually is
spectrally smooth, but we also need a practical approach for constructing a smooth phase spectrum with a noise level
significantly below the noise level of the complex spectrum. In practice, we achieve this by using only a short section of the
interferogram around ZPD. Thereby, the smooth phase spectrum is set by the equation

$$|s(v)|e^{i\varphi(v)} = \int_{x=-\varepsilon\cdot MPD}^{+\varepsilon\cdot MPD} I(x)e^{-2\pi vx} \cdot A(x)dx \qquad (3)$$

Here, the dimensionless multiplier $\varepsilon$ denotes that only a fraction of the complete interferogram recorded up to MPD is used.
The function $A(x)$ denotes a strong numerical apodization function, as any non-local ringing extending out from a spectral
position with high signal level would disturb the phase in the surrounding spectral region.





We finally need to mention that interferograms might be recorded "single-sided" or "double-sided". Often, when an
interferometer is designed for achieving higher spectral resolution, the symmetry of the design is abandoned. Instead, the
ZPD position shifted near one end of the mechanical scan range, which still needs to be wide enough to reconstruct the phase
spectrum via equation (2), but the high-resolution details are inferred from the single remaining side of the interferogram
which is recorded. Our proposed method can be used in either situation, but it should be noted that in case of single-sided
interferogram recording, the error propagation of a residual phase error is much more critical, as sine contributions do not
cancel out (as one side of the interferogram is missing) [Brault, 1996; Brasunas and Cushman, 1997], so a very accurate
reconstruction is even more relevant in this case.

The reader finds detailed presentations of all the aspects of FTIR spectroscopy shortly summarized above in text books and
articles [Herres and Gronholz, 1985; Davis et al., 2001; Griffiths et al., 2007].

In section 2, we present the types of spectrometers we used to test the proposed phase correction method. Section 3 describes
a robust scheme for phase unwrapping and the fitting procedure for retrieving the parameters of the phase model. Section 4
investigates the characteristics of phase spectra for the spectrometers introduced in section 2.
**2 Materials and Methods**
This work has been performed in the framework of the FRM4GHG project (Fiducial Reference Measurements for
Greenhouse Gases; https://frm4ghg.aeronomie.be/) supported by European Space Agency (ESA) [Sha et al., 2020]. In the
framework of this project, among further topics related to fiducial reference measurements (FRM), the adequacy of different
portable spectrometers is investigated. Interferograms recorded with these spectrometers have been used for testing the
proposed phase reconstruction algorithm. We shortly present these spectrometers in the following.

The EM27/SUN Fourier-transform spectrometer (FTS) prototype has been developed by Karlsruhe Institute of Technology
(KIT) in cooperation with Bruker Optics, a well-known manufacturer of FTIR spectrometers. It uses a folded pendulum-
corner cube interferometer ("RockSolid" ® design) and employs two room temperature InGaAs detectors to cover the near-
infrared range from $4000 - 12\,000$ cm$^{-1}$. A solar tracker using Camtracker active feedback to control the position of the solar
image on the fieldstop of the spectrometer is directly attached to the spectrometer [Gisi et al., 2011]. Further instrumental
details of the EM27/SUN FTS design characteristics are provided by Gisi et al. (2012) and Hase et al. (2016). Since 2014,
the EM27/SUN FTS is available from Bruker as a commercial item. Meanwhile, more than hundred units are sold and are
operated worldwide by various working groups for atmospheric greenhouse gas measurements; they are especially suited for
the quantification of local sources as cities [Hase et al., 2015], coal mines [Luther et al., 2019; Luther et al., 2022], oil and
gas production areas [Kille et al., 2019], and landfills [Tu et al., 2022]. As an operational framework for guaranteeing





common instrumental and data analysis standards among the operators, the COCCON has been established since [Frey et al.,
2019; Alberti et al., 2022], which is significantly supported by ESA through FRM4GHG and further contracts.

The Bruker IRcube or "Matrix" FTIR is a compact OEM instrument operating in the mid or near IR regions and configurable
for a wide range of laboratory and industrial applications using a range of sampling accessories. In its basic form it contains
a folded pendulum-corner cube interferometer similar to the EM27/SUN ("RockSolid" ® design) with 25mm beam diameter
and either 1 cm$^{-1}$ double-sided or 0.5 cm$^{-1}$ single-sided resolution. As used at the University of Wollongong for solar
measurements, the IRcube includes a source module which accepts a focussed input beam into a selectable aperture (the field
stop) and collimates it, the interferometer, and detector optics module focussing the parallel beam exiting the interferometer
onto a 1mm InGaAs detector via a short focal length off axis paraboloidal mirror. The solar beam is collected from a solar-
tracker-mounted telescope via a 20 m optical fibre – the beam exiting the fibre is focussed into the field stop of the IRcube's
source module.

The Vertex70 spectrometer is produced and sold commercially by Bruker Optics. It was recently replaced in Bruker's
production line by a successor named Invenio. One Vertex70 FTS was purchased in the framework of the FRM4GHG
campaign to be tested alongside the EM27/SUN and IRCube with the reference IFS125HR and AirCore measurements. The
Royal Belgian Institute of Space Aeronomy (BIRA-IASB) and the University of Bremen (UB) performed minor
modifications to the optical components of the Vertex70 and coupled it with a solar tracker to perform solar absorption
measurements. The feasibility to accommodate two detectors (InGaAs and InSb) in the spectrometer allows covering
simultaneously the near- and mid-infrared (NIR and MIR) spectral regions. The measured spectra are analysed to retrieve
column abundances of $XCO_2$, $XCH_4$, XCO and $XH_2O$ in the NIR spectral region and column abundances of methane ($CH_4$),
nitrous oxide ($N_2O$), formaldehyde (HCHO) and carbonyl sulphide (OCS) in the MIR spectral region are currently studied
[Zhou et al., 2023; Sha et al., 2024]. The spectrometer showed comparable results for the retrieved trace gases as those
retrieved with the high spectral resolution FTIR spectrometers. An automated enclosure system has been developed to
deploy the spectrometers autonomously in the field and enhance the coverage of the fiducial reference FTIR data. The aim is
also to use it in future as a traveling standard improving consistency among FTIR data taken at different sites in the MIR
spectral region. The NIR retrieved target gases are part of the COCCON while the data retrieved in the MIR spectral range
can complement the NDACC-FTIR data. This activity is supported by ESA through FRM4GHG contracts.

The Izaña Observatory (IZO) is a high-mountain station located on the island of Tenerife (Canary Islands, Spain) in the
subtropical North Atlantic Ocean (28.3ºN, 16.5ºW) at an altitude of 2.37 km a.s.l. IZO is managed by the Izaña Atmospheric
Research Centre (IARC, https://izana.aemet.es/, last access: 5 August 2024), which belongs to the State Meteorological
Agency of Spain (AEMet). Within the IZO's atmospheric research activities, the FTIR programme started in 1999 in the
framework of a collaboration between AEMET and KIT [Schneider et al., 2005], contributing to NDACC and TCCON



networks since 1999 and 2007, respectively. To do so, the IZO FTIR instrument, currently a Bruker IFS125HR based on a
Michelson interferometer, records high-resolution solar absorption spectra in the MIR region within NDACC activities and
in the NIR region for TCCON retrievals, using a set of different field stops, narrow-bandpass filters, and detectors
[Schneider etThis  al., 2010; García et al., 2021].

For TCCON the IFS125HR FTIR measures between 4000 and 10,000 cm$^{-1}$ at a spectral resolution of 0.02 cm$^{-1}$ (MPD of 45
cm) using a calcium fluoride (CaF$_2$) beamsplitter, an extended Indium–Gallium–Arsenide (InGaAs) photodiode detector
operated at room temperature, and no optical filters. The operational TCCON spectra are the result of co-adding six single-
sided interferograms in order to increase the signal-to-noise ratio. These interferograms are acquired with a scanner velocity
of 20 kHz, so the acquisition of one solar spectrum lasts about 4 minutes.

**3 New Phase reconstruction scheme**
The drawback of the classical method described in the introduction is twofold. (1) The reduction of the phase spectrum to the
desired very low resolution is achieved explicitly by using a very short section of the interferogram around ZPD for the
Fourier transform [Mertz, 1965; Forman et al., 1966]. This approach neglects interferogram data points further out which
still could contribute information on the phase. (2) The resulting spectral interpolation as part of the procedure is not well-
defined especially across spectral sub-regions of increased opacity, as they occur in solar absorption spectroscopy between
the atmospheric window regions and in strong absorption bands. Because the phase spectrum across such a region is strongly
impacted by the overlapping contributions to the phase emerging from either side of the opaque region, the outcome for the
phase at a certain spectral position in the region with reduced transmission will depend on the user-selected resolution for the
phase calculation and the chosen apodization function.

We will achieve our enhanced reconstruction of the phase spectrum by fitting a smooth parameterized phase model through a
calculated phase spectrum, which preserves higher spectral resolution than required for the desired degree of spectral
smoothness. The smoothness of the phase spectrum is ensured by the phase model used, while avoiding the aforementioned
problems of the classical method. We use a least squares fit of the model to the raw phase spectrum, which is a well-defined
process with respect to interpolation. A similar method has been proposed by Learner et al., 1995, in the context of emission
spectra.

The phase spectrum is a function of angular orientation, so it is invariant under phase shifts of size $\pm 2\pi n$, with $n = 1,2,3, ....$
For our fit procedure, we need to ensure that the raw phase used as input does not include jumps between such branches. We
suggest the very robust procedure summarized as procedural steps in Table 1.





This proposed method can fail if the phase difference calculated in step 5 is greater than ±π. We did not encounter this
situation, but it may occur if the phase slope is very steep and can possibly be avoided by appropriate repositioning of the
ZPD point when calculating the Fourier Transform.
Table 1: step-by-step procedure for developing the raw phase used as input for the model fit.

| Step # | Procedure | Comment |
|---|---|---|
| 0 | Allocation of arrays:<br>(1) float array for accepting phase values<br>(2) logical array indicating availability of valid phase value | |
| 1 | Establish the noise level and the size of potential artefacts superimposed on the spectral signal. Set a threshold $T$ for the subsequent phase calculation significantly above noise and artefact levels. | |
| 2 | Search for position of max amplitude of $s(v_i)$ in the complex signal in the optical bandpass. | Restrict search to relevant optical bandpass, as out-of-band artefacts triggered by source brightness fluctuation might create very big amplitudes at $v \approx 0$. |
| 3 | Calculate phase at spectral index $istart$ with max signal amplitude. | Use a quadrant-sensitive $atan2$ function on real and imaginary part of the complex signal. |
| 4 | Move from current position one spectral index up. If still within the defined spectral bandwidth, check whether $s(v_i) > T$. If so, set logical array value of current position to $true$, otherwise to $false$. | |
| 5 | If the logical array value of the current position $i$ is $true$, calculate the phase difference between the nearest preceding point $j$ assigned $true$ and the current position. | Use the value of the cross product between the normalized complex pointers:<br>$$\Delta\varphi(j \to i) = asin\left\{\frac{\left((s(v_j) \times s(v_i))\right)}{|s(v_j)||s(v_i)|}\right\}$$ |
| 6 | Calculate the new phase value at the current position using the phase value of the nearest preceding point | $$\varphi(i) = \varphi(j) + \Delta\varphi(j \to i)$$ |
| 7 | Continue steps 4 + 5 + 6 until the upper limit of the spectral bandwidth is reached. | |





| 8 | Return to position *istart* and use the corresponding procedure in downwards direction until the lower limit of the spectral bandwidth is reached. | |

The second step is to fit the parameters of the phase model to the raw phase values. We assume here use of a model linear in the model parameters to be fitted. However, nonlinear models also can be handled in our approach by implementing an iterative search for the optimal model parameter values. If a sophisticated model is chosen, which intends to describe actual physical characteristics of the spectrometer (dispersion curves, electronic response characteristics) and retrieves physical quantities (layer thicknesses, capacitances, resistor values), using a model which is nonlinear in the parameters might be unavoidable. When constructing ad-hoc models which simply enforce smoothness, the choice of a simple linear model seems advisable. The fitting procedure needs to be restricted to points for which valid phase values were established in the previous step. The fitting procedure can take into account a weighting according to the squared signal amplitude. We found very little effect of including this refinement in the determination of model parameters, so we did not implement it in the current pre-processing scheme. Taking a weighting into account, the equation for fitting the phase model parameters becomes

$$\vec{p}_{model} = (K^T W K)^{-1} K^T W \vec{\varphi}_{raw} \tag{4}$$

Here, $\vec{p}_{model}$ is the set of model parameters, $K$ is the Jacobean matrix, which holds the derivatives of the phase model at each spectral grid point with valid raw phase entry, $W$ is a diagonal matrix with $\frac{1}{(s(v_i))^2}$ entries (again, for each spectral grid point with valid raw phase entry), and $\vec{\varphi}_{raw}$ is the vector containing all valid raw phases. Note that the vector dimension of $\vec{\varphi}_{raw}$ and $\vec{p}_{model}$ differ, as after receiving the set of model parameters, $\vec{p}_{model}$ can be calculated at all spectral positions, including interpolation across near opaque spectral sections and extrapolation beyond the first or last spectral point found in the optical bandpass. The predicted model phase values further outside of the relevant spectral bandpass are meaningless and might be suppressed altogether (by allocating the array for $\vec{p}_{model}$ to fit the relevant spectral bandpass).

**4 Results**

For the actual work on the FTIR spectrometers introduced in section 2, we used a polynomial model of order 7. The raw phase calculation uses a resolution of about 10 $cm^{-1}$, which is supported by all spectrometers we included in the study (sufficient number of points on the short side of the interferogram).



## 4.1 Phase spectrum of the EM27/SUN FTS

The results achieved for the EM27/SUN are shown in Figure 1. The spectrometer shows a remarkably linear phase spectrum across the whole spectral region of the main detector (covering 5000 to 12000 cm$^{-1}$). The differences between the model fit and the raw phase are below 1 mrad. The level of smoothness and linearity of the phase spectrum is outstanding among all spectrometers tested. This behaviour probably is supported by the beamsplitter design. The same optical plate is passed twice by the radiation, acting as substrate of the beam-splitting coating layer in one passage and as compensating plate in the other passage. In addition to this, also the analogue electronic chain seems to introduce only minimal dispersion due to runtime effects. It is not clear why the other spectrometers investigated here, all built by the same manufacturer, show significantly stronger structures in the phase spectrum.

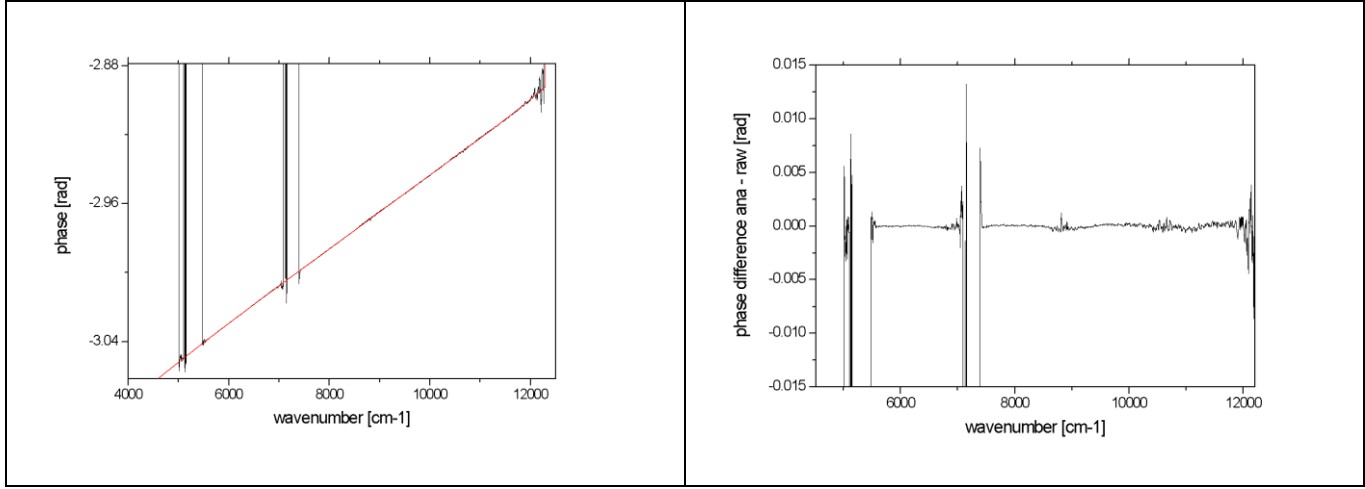

Figure 1: EM27/SUN phase spectrum. Left panel: raw phase (black) and fitted model (red). Right panel: difference between model (analytical, ana) and raw phase (raw). The gaps in the raw phase are due to opaque spectral sections.

## 4.2 Phase spectrum of the IRCube FTS

The phase spectrum of the IRCube is shown in Figure 2. The spectral bandpass covers the range of 4000 to beyond 12 000 cm$^{-1}$. The differences between the phase model and the raw phase show more structure than in case of the EM27/SUN, but still, these oscillatory features are largely within 2 mrad.



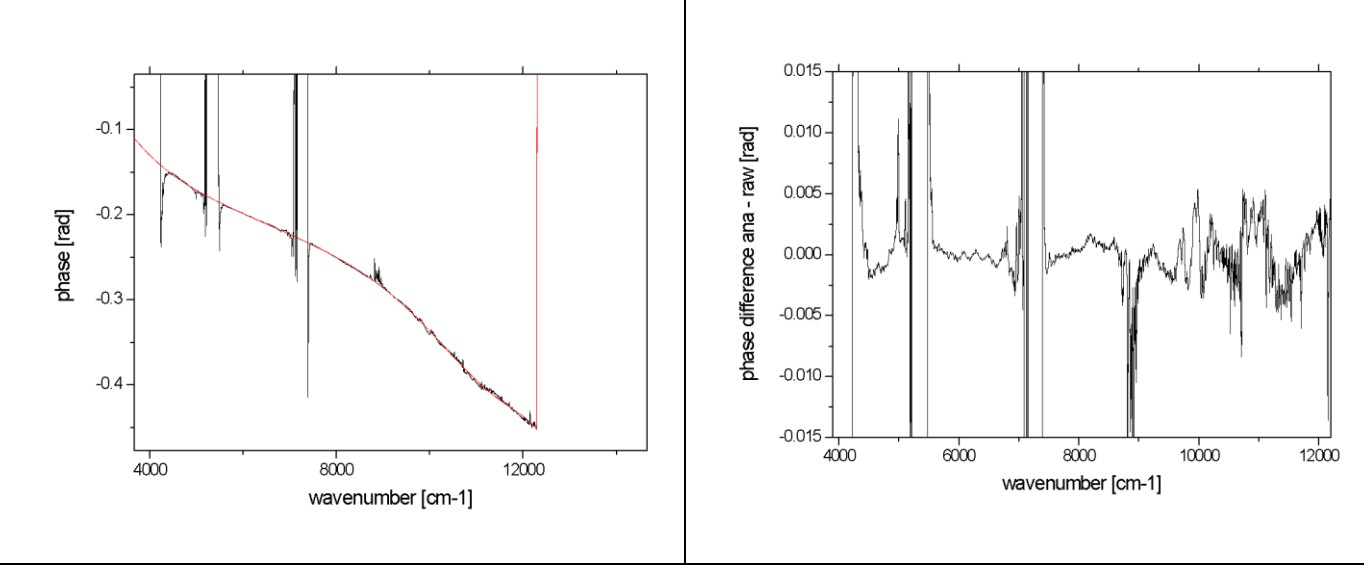

Figure 2: IRCube phase spectrum. Left panel: raw phase (black) and fitted model (red). Right panel: difference between model (analytical, ana) and raw phase (raw). The gaps in the raw phase are due to opaque spectral sections.

**4.3 Phase spectrum of the IFS125HR FTS operated at Izaña**

The phase spectrum of the IFS125HR operated at the Izaña observatory is shown in Figure 3. The spectral bandpass covers 4000 to beyond 12 000 cm$^{-1}$. Due to the facts that Izaña is a high-altitude site and a low threshold value for the phase calculation was used because of the very low noise level of the measurements, there are no gaps in the raw phase. Some structure can be seen in the model minus raw phase difference, but this is still within mostly 2 mrad apart from the highest wavenumbers. The curvature of the phase is somewhat stronger than in case of the IRCube. The sharp peaks occurring around 5400 and 7200 cm$^{-1}$ are coinciding with near-opaque regions of the spectrum and might hint at superimposed spurious signals, potentially due to residual nonlinearity. Such spurious signals generally possess a phase orientation different from the real signal. This finding demonstrates that the model-fitting approach presented here might also be useful for detecting different kinds of imperfections in measured spectra.





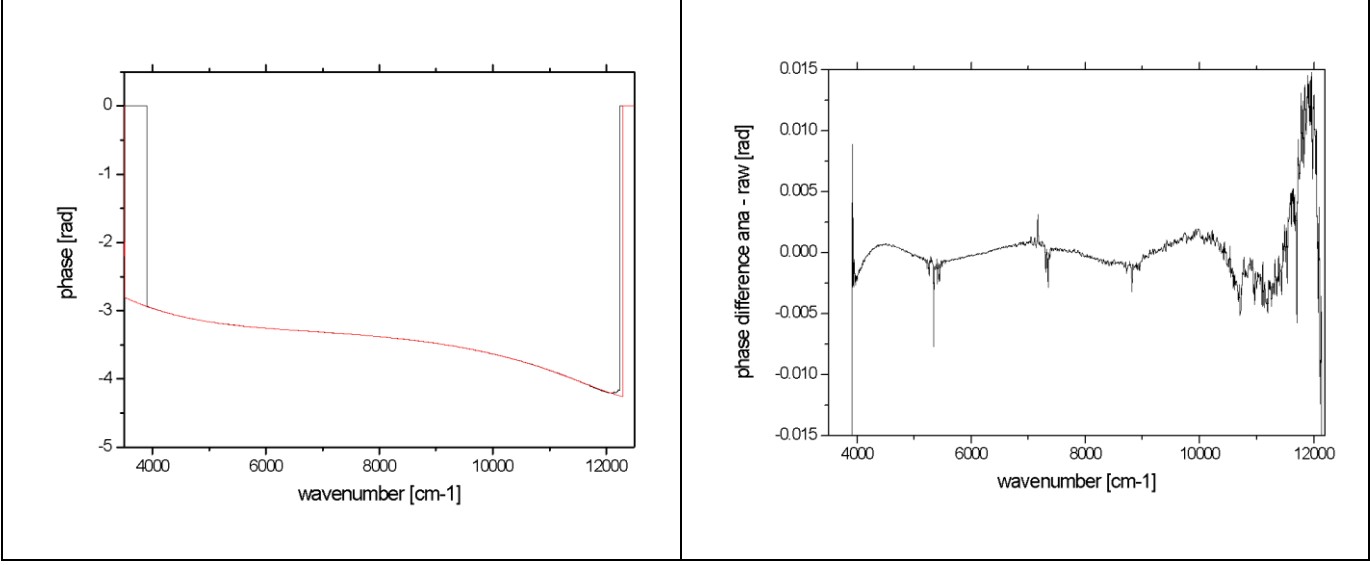

Figure 3: IFS125HR phase spectrum. Left panel: raw phase (black) and fitted model (red). Right panel: difference between model (analytical, ana) and raw phase (raw).

## 4.4 Phase spectrum of the Vertex 70 FTS

Figure 4 shows the phase spectrum of the Vertex 70 FTS. The spectral range covered extends from around 4000 to beyond 12 000 cm$^{-1}$. It is the most unusual phase spectrum we found, showing pronounced quasi-periodic oscillations of about 600 cm$^{-1}$ cycle length in the raw phase (see right panel), which cannot be fitted by the polynomial model used. The amplitude of these oscillations amounts to $\pm 5$ mrad. A very similar oscillatory structure is present in the successor of this spectrometer offered by Bruker under the model name Invenio (not shown here). We reported back our findings to the manufacturer, but so far no explanation or remedy for the unusual behaviour was found. Again, it turns out that the approach presented here to fit a smooth model phase to the raw phase is useful for discovering such instrumental characteristics which otherwise remain overlooked. If the approach presented here is to be applied in an operational way for Invenio measurements, a specific model extension must be designed that allows to reproduce the oscillatory features found in the raw phase.





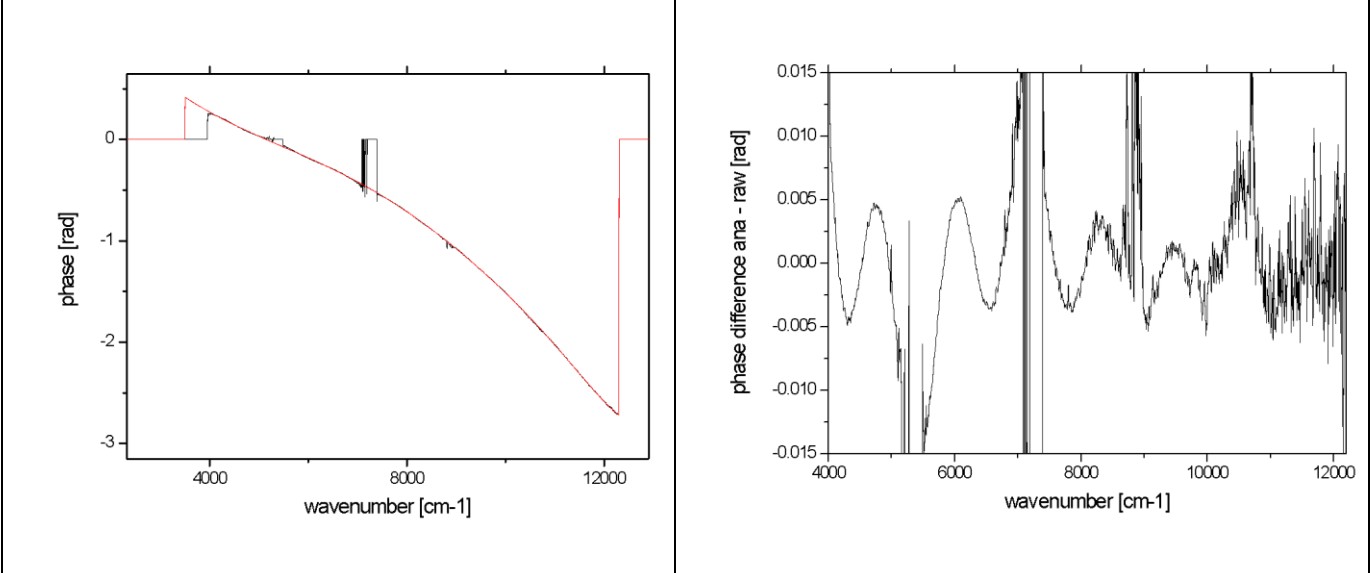

Figure 4: Vertex 70 phase spectrum. Left panel: raw phase (black) and fitted model (red). Right panel: difference between model and raw phase. The gaps in the raw phase are due to opaque spectral sections.

## 5 Impact of the phase on the spectrum and on retrieved gas columns

Figure 5 shows the effect of using either the Mertz or the analytical phase when calculating the spectrum from the measured interferogram. We here use the EM27/SUN and the IRcube for illustration and we investigate the spectral region used for the analysis of $CO_2$ (~ 6200 – 6400 cm$^{-1}$). The EM27/SUN phase spectrum is nearest to a straight line, and the differences between Mertz and analytical phase are well within 1 mrad in the $CO_2$ region (see Figure 1). The IRcube phase spectrum has stronger curvature, but the model used for the analytical phase still delivers a good fit. The differences between Mertz and analytical phase are mostly within 2 mrad in the $CO_2$ region (see Figure 2).

According to Figure 5, the spectral differences of the IRcube spectra are significantly larger than for the EM27/SUN. This reminds of the fact that double-sided interferogram recording has an important intrinsic advantage over single-sided interferograms, because the propagation of a phase error into the spectrum is much more critical for single-sided interferograms. While sine contributions emerging from $\pm OPD$ cancel out in double-sided interferograms, they give rise to point-symmetric residuals around spectral lines in spectra generated from single-sided interferograms. Securing an optimized





phase reconstruction is of higher importance for single-sided interferograms (all the spectrometers investigated here apart
from the EM27/SUN) than for the EM27/SUN, which essentially is insensitive to phase errors in reasonable limits.

The spectral residuals found for the IRcube are quite moderate (below the $10^{-4}$ level), on the other hand both the increasing
demands to be met for the validation of new space borne GHG missions as well as the desired ability to quantify local
sources from differential column measurements make $XCO_2$ measurements with accuracies in the 0.05 ppm range desirable
($\sim 10^{-4}$).

The analysis of the IRcube spectra indicates a change of $CO_2$ column of about $2 \cdot 10^{-5}$ between the two phase corrections
methods, which is not expected to dominate the IRcube error budget. But other tested spectrometer types showed more
pronounced spectral structures in the phase (factor two to five higher amplitudes), which are not negligible.

In general, there is no guarantee that the analytical phase solution is nearer to the truth than the Mertz phase spectrum. The
results always need to be evaluated in context of the specific application. The analytical model might require extensions to
include unexpected phase oscillations (as for the Vertex 70). In any case, however, the analytical method is highly useful to
carve out unexpected structures in the Mertz phase, which are easily overlooked without performing a comparison to the
smooth analytical phase. A careful analysis of such features might help to further improve the design of interferometers and
supports recognition of instrumental problems, because the non-local spectral artefacts created by various error sources (as
nonlinearity, sampling ghosts, double passing) also create disturbances of the phase spectrum.

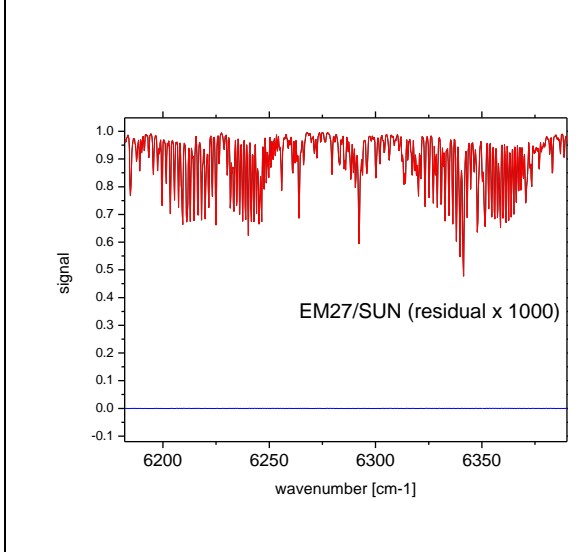
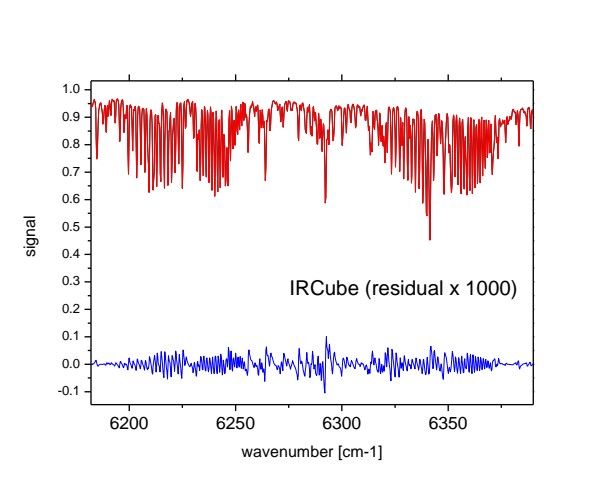






Figure 5: differences of spectra as resulting from the Mertz phase correction scheme and the analytical phase approach. Left: EM27/SUN, right: IRcube, the spectral residuals are enlarged by a factor of 1000.

**6 Summary and Conclusion**

We have implemented a refined method for reconstructing the phase spectrum of FTIR spectrometers. We have applied the new method to different types of spectrometers and found pronounced differences in phase imperfections between them. Our findings demonstrate the usefulness of the method proposed both for operational work and instrumental diagnosis. The proposed algorithm has been incorporated in the COCCON pre-processing code, which is available under the GNU General Public License version 3.

**Authors Share**

FH has implemented the new method for phase correction using analytical model fits of the phase spectrum. He has generated the results for the different spectrometers investigated in this work and wrote the predominant part of the manuscript. All authors have studied and commented on the manuscript.

**Competing Interests**

At least one of the (co-)authors is a member of the editorial board of Atmospheric Measurement Techniques.

**Acknowledgements**

The establishment of the Izaña TCCON site was supported by grants from NASA's terrestrial carbon cycle program and from the OCO project office.

**Financial Support**

This research has been supported by the European Space Agency (contract 400136108/21/I-DT-lr).

**Code Availability**

The COCCON software suite including the pre-processing software PREPROCESS is made available under GPL version 3 license. From version 2.3 onwards, it supports the option of using the analytical phase model implemented in



PREPROCESS. The software suite and source codes are available for download at https://www.imk-
asf.kit.edu/english/3225.php.

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
