# Peer review of "Implementation and application of an improved phase spectrum"

_Atmospheric Measurement Techniques, 2024_

## Referee Comment (RC2)

Hase Phase Correction Review

General Comments.
This is a very good paper describing a new method of phase correction, which is an important step in the processing of FTIR interferogram data into spectra. Inadequately performed phase correction can cause serious artifacts in the resulting spectra, leading to degraded accuracy of the atmospheric gas amounts retrieved from them, especially in spectral regions containing blacked-out absorption lines. As the accuracy required for atmospheric gas measurements gets more and more stringent, it is very timely that the phase correction process be re-examined and upgraded.

The paper has very high standard of English, especially considering that the authors are all non-native English speakers. I only found a handful of instances where the text needed slight adjustments.

Paper contains too much material describing Bruker spectrometers and the Izana site. This interrupts the thread of the science discussion. Individual instances are cited below in the Specific Comments section below.

Paper ends rather abruptly, with no discussion of the impact on retrieved xCO2 of the new phase correction for the Vertex or IFS125 spectrometers.

"phase unwrapping" is mentioned in the Abstract and Introduction, but I found no further discussion of this topic.

The effect of the improved phase correction algorithm is very small, at least in the CO2 bands used by TCCON/COCCON at 6160 to 6380 cm-1, but this window rarely saturates. Perhaps the authors should also investigate a spectral window that contains saturated spectral features close to absorption lines of interest (e.g. HF at 4039 $cm^{-1}$, CO at 4233 or 4290 $cm^{-1}$) since it is under these partially saturated conditions where the new algorithm is supposedly most beneficial. I think that it is a serious omission for the paper not to have investigated some more adverse fitting windows, in which the advantages of the new method would be more apparent.

Specific/Technical Comments (Authors' words in black. My comments in blue.)

Lines 30-33: The word "shortly" usually connotes time, in which context it means "soon". So, I suggest changing sentence to: "Fourier Transform Spectrometry is an important technique for remote observation of atmospheric composition, especially in the near and mid infrared spectral regions, where it is mostly referred to as Fourier Transform Infra-Red (shortened to FTIR) spectroscopy. "
* * *
Lines 38-42: ATMOS should be mentioned here; the first high-resolution FTIR spectrometer to fly in space. Farmer, Crofton B. (1987), High resolution infrared spectroscopy of the sun and the earth's atmosphere from space. Mikrochimica Acta, 93. 189-214 doi:10.1007/bf01201690
* * *
Lines 17 & 19: The authors use the word "connect" to express the relationship between the interferogram domain and the spectral domain. For example, it is used twice in the first four lines of the abstract: "…for concluding which spectral distribution **connects** with the measured interferogram. We present implementation of an improved scheme for phase determination in the operational Collaborative Carbon Column Observing Network (COCCON) processor. We introduce a robust unwrapping scheme for retrieving a **connected** phase spectrum….". To me, "connected" is too vague term. I'm not sure what it is supposed to convey. So, I suggest replacing the first "connects" and deleting the second as follows:

"…for concluding which spectral distribution **most likely gave rise to** the measured interferogram. We present implementation of an improved scheme for phase determination in the operational Collaborative Carbon Column Observing Network (COCCON) processor. We introduce a robust unwrapping scheme for retrieving a phase spectrum…."
* * *
Line 21: "suited" → "suitable"?
* * *
Line 31: Here you use the term Fourier Transform Spectrometry, which I believe is correct. In other places you speak of absorption "spectroscopy", e.g., lines 62-70 contain 4 instances. Are you using the words "spectroscopy" and "spectrometry" interchangeably, or are you making a subtle distinction? If the latter, please explain in the paper. IMO all instances should be "spectrometry" since "spectroscopy" is the study of the interaction of electromagnetic radiation with matter, which is not what TCCON or COCCON do.
* * *
Line 73: "maximum optical path difference (MPD) is …." Why doesn't "optical" participate in the acronym (MOPD)?
* * *
Lines 75-76: "A proper description of the instrumental line shape (ILS) is further complicated due to the presence of practical imperfections of the interferometer. "
Need to give an example or two of these "practical imperfections", or add "as will be shown later". Otherwise, the reader is left hanging.
* * *
Line 90: You don't mention the compensator here. Surely, the mismatch in thickness and/or refractive index between the beamsplitter and compensator is a major cause of phase error.
* * *
Line 97: The term S(v) multiplies the integral in equation (1), but here in eqtn (2) it is equal to the integral. Is there a missing "=" in eqtn. (1)?
* * *
Line 97: Earlier (line 74) it was mentioned that the interferogram is multiplied by a boxcar function. Why is this not shown in eqtn (2)?
* * *
Line 107: "The assumption of uncorrelated white noise typically is adequate". Adequate for what purpose?
* * *
Line 111: "The assumption of a spectrally smooth phase allows to separate at each spectral position the complex spectrum into two orthogonal components"
"allows to separate" is a construction that is not used in English speaking world. I suggest re-writing as: "The assumption of a spectrally smooth phase allows separation of the complex spectrum into two orthogonal components"
* * *
Line 112: Please clarify what you mean by "direction".
* * *
Line 131: "Instead, the ZPD position shifted near one end of the mechanical scan range"
Re-write as "Instead, the ZPD position is shifted to be near one end of the mechanical scan range"
* * *
Line 132: I think that "equation (2)" should be "equation (3)".
* * *
Lines 151 to 188: This section interrupts the flow of the scientific discussion, by providing a lot of mostly irrelevant information about the various Bruker spectrometers. For example, the fact that they use the "Rock-Solid" design is mentioned twice here, as is the fact that "more than 100 units are sold". This

section should be considerably shortened, or moved to an appendix. Perhaps cite the Bruker website/brochure for readers who want this type of information.
* * *
Lines 191-195: "IZO is managed by the Izaña Atmospheric Research Centre (IARC, https://izana.aemet.es/, last access: 5 August 2024), which belongs to the State Meteorological Agency of Spain (AEMet). Within the IZO's atmospheric research activities, the FTIR programme started in 1999 in the framework of a collaboration between AEMET and KIT [Schneider et al., 2005], contributing to NDACC and TCCON networks since 1999 and 2007, respectively."

I don't find this mostly historical information scientifically relevant. Which organizations manage and fund the IZO site, and how long they have done so, are not relevant to phase correction. Most of this should be moved to the Acknowledgements. The website https://izana.aemet.es contains nothing of relevance to phase correction. Also, the paragraph doesn't mention COCCON, which I thought was the main driver of the new phase correction method.
* * *
Line 170: "1mm" → "1 mm". In most places a space is left between the value and the unit, but here not.
* * *
Line 198: "Schneider etThis al. " → "Schneider et al."
* * *
Lines 202-215: "Because the phase spectrum across such a region is strongly impacted by the overlapping contributions to the phase emerging from either side of the opaque region, the outcome for the phase at a certain spectral position in the region with reduced transmission will depend on the user-selected resolution for the phase calculation and the chosen apodization function."

I don't dispute this statement – this is what you found. But it doesn't explain why the phase correction operator is so sensitive to the user-selected resolution or apodization. The exact phase value will of course have some dependence on the interferogram points selected (i.e. the phase resolution) and their relative weighting (i.e., apodization). But why does this have a large effect on the phase?

Also, It seems contradictory that points far from ZPD contain some information about the phase, that you wish to retain, but you subsequently fit a low-order polynomial through the phase, which is smoothe, and therefore contains no high-resolution frequencies coming from igram points far from ZPD.
* * *
Lines 227-229: "This proposed method can fail if the phase difference calculated in step 5 is greater than $\pm\pi$. We did not encounter this situation, but it may occur if the phase slope is very steep and can possibly be avoided by appropriate repositioning of the ZPD point when calculating the Fourier Transform.

This is of concern because when automatically processing thousands of interferograms, how would we know when the ZPD point needs to be repositioned?
* * *
Table 1: "logical array indicating availability of valid phase value " → "logical array indicating validity of phase value"
* * *
Table 1: Why isn't $s(v_i)$ allocated in step 0?
* * *
Table 1: Perhaps give names to the float array containing phase and the logical array containing validity e.g. Psi and LVALID. This would avoid repeating "logical array value of current position" later in the table (twice).
* * *
Table 1: "Use the value of the cross product between the normalized complex pointers"

What are "normalized complex pointers"? There is no mention of "pointers" anywhere else in the paper.

Table 1: At which step is index $j$ updated? Somewhere it is set to $i$, before $i$ is incremented.
* * *
The equation $\Delta\varphi_{(j \to i)} = \text{asin}\{\}$ in Table 2 finds the phase difference between the complex vectors $Sv_i$ and $Sv_j$ by using their cross product. It is not obvious why this is the case. To me, it would be simpler to directly subtract the phases at $v_i$ and $v_j$ as follows:

Let $Sv_i = [a+ib]$ , $Sv_j = [c+id]$, where i=Sqrt(-1). The phase of $Sv_i$ is atan[b/a] and that of $Sv_j$ is atan[d/c].

$\Delta\varphi_{(j \to i)} = \varphi_i - \varphi_j = \text{atan}[b/a] - \text{atan}[d/c] = \text{atan}[(ad-bc)/(ac+bd)]$

To me this approach seems more intuitive than using cross-products. So please explain the advantage using a vector product, rather than simply subtracting the two phases.
* * *
Table 1: Where does the "phase unwrapping" occur? This is mentioned earlier in the paper including the abstract. But then nothing more.
* * *
Line 226: Here Table 1 is mentioned, but there is no description in the main body of text. So the table itself and its caption need to be more self-explanatory.
* * *
Line 234: "The second step…". At first, I thought that this referred to Table 1 (Step #). But that doesn't make sense. So perhaps change this sentence to "After completing the raw phase vector over the full bandwidth, as described in Table 1, we next fit the parameters of the phase model to the raw phase values."
* * *
Line 247: Need to clarify whether P$model$ is the retrieved model parameters (e.g., polynomial coeffs) or the reconstructed phase values. Line 247 states " P$model$ is the set of model parameters". But line 250 states "after receiving the set of model parameters, P$model$ can be calculated at all spectral positions", which suggests that P$model$ is the reconstructed phase values themselves.
* * *
Line 256: "phase calculation uses a resolution of about 10 $cm{-}1$, which is supported by all spectrometers we included in the study (sufficient number of points on the short side of the interferogram)." It would be helpful to some readers to also express this as interferogram points. Assuming the usual two samples per reference laser wavelength, I think that 10 cm$^{-1}$ corresponds to a MOPD of 0.9/10=0.09 cm which is 0.0/0.3154E-04=2844 interferogram points on each side of ZPD. If correct, perhaps state this number.

Figure 1: Why does the right-hand panel have a slightly narrower wavenumber range that the left-hand panel, or figs. 2, 3, 4.
* * *
Line 269: "model (analytical, ana)". What is "ana"? I puzzled over this for a while before deciding that you are defining a new abbreviation. But why not simply label the y-axes of the figs "analytic - raw" – there's plenty of room. The "ana" abbreviation isn't used anywhere else.
* * *
Line 269: Is the "fitted model (red)" the same thing as "model (analytical, ana)"?  If so, use the same terminology. If they are different, explain the distinction.
* * *
Line 283: "The curvature of the phase is somewhat stronger than in case of the IRCube." →
"The curvature of the phase is somewhat stronger than in **the** case of the IRCube."
* * *
Line 308: "Figure 5 shows the effect of using either the Mertz or the analytical phase " → "Figure 5 shows the effect of using either the **classical** Mertz or the analytical phase…"
* * *
Line 316: "This reminds of the fact that…" → "This is a reminder of the fact that"
* * *
Line 320: "…phase reconstruction is of higher importance for single-sided interferograms (all the spectrometers investigated here apart from the EM27/SUN) than for the EM27/SUN, which essentially is insensitive to phase errors in reasonable limits" This is a bit clumsy. I suggest re-writing as: "…phase

reconstruction is much easier and accurate for double-sided interferograms (i.e., the EM27/SUN) than for single-sided interferograms (the other spectrometers discussed here)"
* * *
Line 323-326: Split into two sentences.
* * *
Line 326: Needs a reference for why atmospheric xCO2 measurements need an accuracy of 0.05 ppm.
* * *
Line 328: Is $2 \cdot 10^{-5}$ a fractional change?  So this is a 425ppm*$2 \cdot 10^{-5}$ = 0.0085 ppm change in xCO2. This is very small. So for the EM-27 or the IR cube, the classical Mertz phase correction is perfectly adequate.
* * *
Line 333: How much difference between the two phase correction methods for the Vertex and IFS125? These instruments seem to have been dropped from the discussion.

---

## Author Response (AR1)

The article "Implementation and application of an improved phase spectrum determination scheme for Fourier Transform Spectrometry" by Hase et. al. describes a more robust method for determining the phase spectrum of a Fourier transform spectrometer and applies the described method to a selection of instruments used for ground-based measurements of atmospheric trace gases.

This work has been carried out in the context of the European Space Agency's Fiducial Reference Measurements for Greenhouse Gases (FRM4GHG) programme and so there is an expected focus on retrieval of greenhouse gases from absorption spectra. The presented results show that the changes in retrieved CO2 column using the new and traditional phase determination are very modest. However, the new and more rigorous method presented can provide insight into the instrumental features of the various spectrometers presented.

We sincerely thank the anonymous reviewer for the careful evaluation of our draft and the very useful resulting suggestions repeated below. In between, we insert our replies in red and sections of the revised draft in green colour.

The manuscript is generally well written, especially the introduction section, and the content will form a beneficial addition to the field. I would recommend that it is published subject to the changes and clarifications outlined below.

Specific comments:

Given that this work was carried out as part of FRM4GHG, the discussion of the impact of the phase determination on retrieved gas column is very limited, with quantified results only presented for one instrument and one species. It would be good to include these results for all of the instruments investigated and multiple species or spectral windows. Even if all the differences are at or below the magnitude reported for the IRcube CO2 window, this is still a useful finding which should be noted.

We agree and have significantly extended section 5. In the updated version, we investigate results covering a wider range of airmasses using IRCube spectra. The $8730 - 8850$ $cm^{-1}$ region and the retrieved XH2O data appears to be an ideal-typical example of a near-opaque band, which shows considerable deviations in the Mertz phase. In our feeling, a systematic investigation for all relevant bands and species is well beyond the scope of this paper, which mainly intends to introduce the algorithms now included in the COCCON preprocessing. (Unfortunately, a systematic investigation of the effects of phase calculation is not covered by the FRM4GHG project. We agree it would be desirable to include a systematic study of phase effects in a follow-up project.)

In section 2 the instruments should be introduced in a more consistent way. For example, there is very little discussion of the location of any of the instruments until the Izaña IFS125HR is introduced, and a paragraph is used to describe the observatory before a second, shorter paragraph describes the instrument.

We have significantly shortened the instrument descriptions and we have added an overview table. We now clarify in the beginning of the section that all portable spectrometers were operated at the Sodankyla site in the framework of the FRM4GHG project.

Furthermore, when describing the IRcube, it is noted that single or double-sided interferograms can be measured, but it is never made explicit which are used in this study.

In order to reach a 0.5 cm$^{-1}$ resolution, the interferometer of the IRcube needs to be configured for single-sided interferograms. If the interferometer is configured for recording double-sided interferogram, then a maximum resolution of only 1.0 cm$^{-1}$ is achievable. However, exclusively the configuration for recording 0.5 cm$^{-1}$ single-sided interferograms is studied in the framework of FRM4GHG (it would be an interesting exercise to test which instrument configuration actually is superior, but this is not covered by the current FRM4GHG project).

A table in this section that summarises the important instrument characteristics would be very helpful to the reader.

We added such a table.

When describing the algorithm for the new phase determination a discussion on the choice of the threshold value T would be useful. What constitutes significantly above the noise and artefact level, and what are the implications of setting this value too low?

The phase unwrapping algorithm requires a locally smooth phase spectrum without sudden changes of phase orientation between adjacent spectral positions reaching or exceeding the value of $\pi$. In order to safely eliminate all points from the process, which might induce ambiguities in the phase unwrapping, a threshold of several standard deviations of the noise level (e.g., 5 or 6) should be chosen. This is not a an overly demanding requirement, as it refers to the noise level of the low-resolution complex spectrum used for the phase calculation, which  has a much lower noise level than the spectrum used for the trace gas analysis derived from the complete interferogram.

The question raised by the reviewer about potential artefacts is of special relevance. Imperfections of the measurement process, as detector nonlinearity, double-passing, or periodic sampling errors, can all superimpose spurious spectral flux to the real signal. This spurious flux is derived from spectral signal located elsewhere and therefore has a differing phase orientation. Contrary to noise superimposed on the phase, the phase orientation of spurious flux is continuous, so it does not vanish when a smooth phase model is fitted through the unwrapped phase spectrum.

Spectrometers used for the remote sensing of GHGs (aiming at accuracies well below the percent level) need to control such artefacts very well. The proposed construction of an analytical phase provides a sensitive tool for detecting residual artefacts in the spectra: when moving along the spectral abscissa into a more opaque region, the spurious flux becomes larger in proportion to the real signal. This introduces a rotation of the unwrapped phase, while the smoother analytical phase largely removes such phase excursions. Therefore, visualization of the difference between unwrapped and the analytical phase allows to recognize spurious flux. However, the proposed method only is a partial cure: while the reconstructed phase in the opaque region will be a much better approximation to the actual phase than the local Mertz phase, the spurious signals are still there and instrumental improvements are required to remove these spurious fluxes.

In Section 3, we have added the following sentences: For generating a phase point of the raw unwrapped phase, the spectral amplitude is required to exceed the adjustable threshold value T. It should be chosen well above the noise level of the complex spectrum used for the phase determination. Otherwise, the phase difference between adjacent points could occasionally exceed the requirement of phase differences to reside within the ±π range. Moreover, the phase in nearly opaque spectral sections can be dominated by spurious signals (originating from, e.g., nonlinearity, double-passing, or sampling ghosts), so it is desirable to exclude these spectral sections from the calculation of the analytical phase anyway.

Technical corrections:

L75. ILS should be more fully introduced

We have included some extensions in this paragraph and added a reference. It now reads (new inserts in boldface):

In equation (1), we have extended the integration over all optical path differences. In practice, only a limited section up to a maximum optical path difference (MPD) is accessible. The truncation of the interferogram is equivalent to a multiplication with a boxcar function. **In spectral domain, this becomes a convolution with a sinc function. The spectral response inherent to an FTIR spectrometer is called instrumental line shape (ILS).** It can be adjusted by applying a numerical weighting function along the interferogram (the process of apodization). **Especially, numerical apodization allows to dampen the sidelobes of the sinc function, which allows – at the cost of widening the ILS width – to suppress the ringing surrounding unresolved spectral lines.** A proper description of the instrumental line shape (ILS) is further complicated due to the presence of practical imperfections of the interferometer **[Hase et al., 1999]**. Finally, we do not further follow the problem of spectral ordinate calibration here, because it, too, is not closely related to our aim of an improved phase reconstruction.

L176. The reference IFS125HR has not been discussed prior to here. Is this the Izaña instrument?

Thanks for rising this point. For clarification, we have added the following statement in the first paragraph of section 2:

For this purpose, extended measurement campaigns with the portable spectrometers under test are performed at the TCCON site Sodankyla operated by the Finnish Meteorological Institute. At this site, also regular aircore measurements are executed, which provide in-situ measurements of Greenhouse Gas profiles. Further details of the campaign setup are provided by Sha et al., 2020.

The reason for using the Izaña spectrometer for demonstrating the phase of an IFS125HR is due to the fact that the required interferograms were at hand (the TCCON data analysis for Izaña is operated by KIT) and that the Izaña spectrometer presents the basic TCCON configuration without optional instrumental extensions or any deviations from the standard setup.

L328. Include units after 2 · 10−5

Thanks, in the discussion of CO column change, we have added the notion "relative change" instead of "change" in order to clarify that the reported change is a unitless quantity.

Figure5. It isn't clear that there are two spectra plotted. Consider including a legend. The curve labelled residual isn't really a residual in the normal sense but a difference, consider relabelling. Include a description of the curve colours in the caption.

In order to clarify what is shown, we changed the ordinate label to "spectral signal" (we hesitate to name it "transmission", as solar absorption measurements lack proper ordinate calibration). We have changed "residuals" to "spectra difference".

General Comments.
This is a very good paper describing a new method of phase correction, which is an important step in the processing of FTIR interferogram data into spectra. Inadequately performed phase correction can cause serious artifacts in the resulting spectra, leading to degraded accuracy of the atmospheric gas amounts retrieved from them, especially in spectral regions containing blacked-out absorption lines. As the accuracy required for atmospheric gas measurements gets more and more stringent, it is very timely that the phase correction process be re-examined and upgraded.
The paper has very high standard of English, especially considering that the authors are all non-native English speakers. I only found a handful of instances where the text needed slight adjustments.
Paper contains too much material describing Bruker spectrometers and the Izana site. This interrupts the thread of the science discussion. Individual instances are cited below in the Specific Comments section below.

We sincerely thank Geoff Toon for the careful evaluation of our draft and the very useful resulting suggestions repeated below. In between those, we insert our replies in red and sections of the revised draft in green color.

Paper ends rather abruptly, with no discussion of the impact on retrieved xCO2 of the new phase correction for the Vertex or IFS125 spectrometers.

We have extended the discussion of impact on retrieved gas amounts. We use an accessible saturated band of $H_2O$ and the IRCube spectrometer for this demonstration (see our comments further below on sensitivities).

"phase unwrapping" is mentioned in the Abstract and Introduction, but I found no further discussion of this topic.

Sorry for the lacking clarity, we missed to provide a clear description. We imagine the proposed method to consist of two partial steps: (1) the construction of a smooth phase spectrum out of the complex spectrum. We refer to this as "phase unwrapping". The algorithm for doing this is outlined in table 1 contained in section 3. (2) The fit of the final analytical phase model to the continuous phase spectrum achieved in the first step. Equation 4 is used for determining the chosen parameters of the phase model.

We have added the following clarification in the manuscript at the beginning of section 3: The method described in the following consists of two partial steps: First, we need to establish a procedure for constructing a smooth phase spectrum from the complex spectrum. We refer to this step as "phase unwrapping". The trigonometric functions connecting phase angle and complex spectrum are periodic, and direct use of inverse functions would generate phase jumps. In the second step, we fit the phase values of an analytical phase model to the smooth phase spectrum generated in the first step by adjusting the chosen model parameters.

In the table description, we added accordingly (in bold): step-by-step procedure **for the phase unwrapping algorithm**, which develops the raw phase used as input for the model fit.

The effect of the improved phase correction algorithm is very small, at least in the CO2 bands used by TCCON/COCCON at 6160 to 6380 cm-1, but this window rarely saturates. Perhaps the authors should also investigate a spectral window that contains saturated spectral features close to absorption lines of interest (e.g. HF at 4039 cm-1, CO at 4233 or 4290 cm-1) since it is under these partially saturated

conditions where the new algorithm is supposedly most beneficial. I think that it is a serious omission for the paper not to have investigated some more adverse fitting windows, in which the advantages of the new method would be more apparent.

We fully agree that the method of phase spectrum reconstruction will be more prominent in spectral sections, which are close to saturation. Unfortunately, for portable low-resolution spectrometers, which are the main focus of FRM4GHG, the strongly under-resolved HF signature is a very difficult target, while the CO window at higher wavenumbers already has significantly higher signal level, so phase determination in this spectral section is not too difficult. The $8730 - 8850$ $cm^{-1}$ spectral region approaches saturation at higher soar zenith angles under humid conditions and inspection of the phase spectrum indicates significant differences between Mertz phase and analytical phase, so it seems a nice test laboratory for phase effects. Therefore, in the revised version of the manuscript, we significantly revised and extended section 5. We demonstrate retrievals of the dominant absorber in this region, $H_2O$, on spectra collected with the IRCube (which delivers phase-sensitive single-sided interferograms) and investigate for a selected day of measurements the changes in the retrieved abundance as function of band saturation.

Specific/Technical Comments (Authors' words in black. My comments in blue.)
Lines 30-33: The word "shortly" usually connotes time, in which context it means "soon". So, I suggest changing sentence to: "Fourier Transform Spectrometry is an important technique for remote observation of atmospheric composition, especially in the near and mid infrared spectral regions, where it is mostly referred to as Fourier Transform Infra-Red (shortened to FTIR) spectroscopy. "

Thanks, we replaced the original sentence!
* * *
Lines 38-42: ATMOS should be mentioned here; the first high-resolution FTIR spectrometer to fly in space. Farmer, Crofton B. (1987), High resolution infrared spectroscopy of the sun and the earth's atmosphere from space. Mikrochimica Acta, 93. 189-214 doi:10.1007/bf01201690

Ok, we added ATMOS in the introduction and in the references!
* * *
Lines 17 & 19: The authors use the word "connect" to express the relationship between the interferogram domain and the spectral domain. For example, it is used twice in the first four lines of the abstract: "…for concluding which spectral distribution **connects** with the measured interferogram. We present implementation of an improved scheme for phase determination in the operational Collaborative Carbon Column Observing Network (COCCON) processor. We introduce a robust unwrapping scheme for retrieving a **connected** phase spectrum….". To me, "connected" is too vague term. I'm not sure what it is supposed to convey. So, I suggest replacing the first "connects" and deleting the second as follows: "…for concluding which spectral distribution **most likely gave rise to** the measured interferogram. We present implementation of an improved scheme for phase determination in the operational Collaborative Carbon Column Observing Network (COCCON) processor. We introduce a robust unwrapping scheme for retrieving a phase spectrum…."

Thanks, we have removed the first two occurrences as suggested. In the third occurrence in the abstract stating that "We introduce a robust unwrapping scheme for retrieving a connected phase spectrum", we changed "connected" to "spectrally smooth".
* * *
Line 21: "suited" ❼ "suitable"?

We changed the expression as proposed.
* * *
Line 31: Here you use the term Fourier Transform Spectrometry, which I believe is correct. In other places you speak of absorption "spectroscopy", e.g., lines 62-70 contain 4 instances. Are you using the words "spectroscopy" and "spectrometry" interchangeably, or are you making a subtle distinction? If the latter, please explain in the paper. IMO all instances should be "spectrometry" since

"spectroscopy" is the study of the interaction of electromagnetic radiation with matter, which is not what TCCON or COCCON do.

We have to admit that the variable use has no further meaning, so we follow your suggestion and consistently changed it to "spectrometry" in the revised manuscript.
* * *
Line 73: "maximum optical path difference (MPD) is …." Why doesn't "optical" participate in the acronym (MOPD)?

We agree this could be done, but because we actually do not need to keep MPD and MOPD apart in the discussion and as we have clarified in the definition that the abbreviation refers to the optical path difference, we would like to maintain the shorter abbreviation, ok?
* * *
Lines 75-76: "A proper description of the instrumental line shape (ILS) is further complicated due to the presence of practical imperfections of the interferometer. "
Need to give an example or two of these "practical imperfections", or add "as will be shown later". Otherwise, the reader is left hanging.

We added two examples and a reference: " …practical imperfections of the interferometer **as misalignment of optical components or mechanical imprecision of the scanning mechanism [Hase et al., 1999]**."
* * *
Line 90: You don't mention the compensator here. Surely, the mismatch in thickness and/or refractive index between the beamsplitter and compensator is a major cause of phase error.

This is true. Our statement "Due to residual optical asymmetry of the beamsplitter …" intended to incorporate this possibility, in the revised manuscript, we now state this more explicitly: "Due to residual optical asymmetry of the beamsplitter **unit (especially due to a potential mismatch of the substrate carrying the beam-splitting layer system and the compensation plate)** …"
* * *
Line 97: The term S(v) multiplies the integral in equation (1), but here in eqtn (2) it is equal to the integral. Is there a missing "=" in eqtn. (1)?

True, somehow the "=" got lost in equation 1, we corrected the error in the revised manuscript. Thanks for spotting!
* * *
Line 97: Earlier (line 74) it was mentioned that the interferogram is multiplied by a boxcar function. Why is this not shown in eqtn (2)?

Good point, thanks for mentioning. The first two equations even hold for the irradiated spectrum (as long as $s(v)$ is identified with the original irradiated spectrum (sometimes called monochromatic or true spectrum)), there is no strict need up to this point of taking into account that in the measurement process the interferogram is truncated to a finite value. We do not think the treatment would become more transparent by explicitly including the truncation. But you are right that in equation 3, it becomes unavoidable to recognize that the complex spectrum is a downgraded version of the irradiated spectrum (and also is downgraded in comparison to the spectrum derived from the complete interferogram). We therefore have extended equation 3 in the revised manuscript:

$$|s(v) \otimes FT(A_{trunc})|e^{i\varphi(v)} = \int_{x=-\varepsilon \cdot MPD}^{+\varepsilon \cdot MPD} I(x)e^{-2\pi vx} \cdot A(x)dx$$

We state that the spectrum is convolved with the truncated apodization function to assure that the equation remains valid even if a function is used which does not fall of to zero at $+-\varepsilon \cdot MPD$.
* * *
Line 107: "The assumption of uncorrelated white noise typically is adequate". Adequate for what purpose?

We have added the statement "The assumption of uncorrelated white noise typically is adequate **for interferogram samples**." The notion just serves for outlining that in spectral domain, the superimposed noise is white noise in FTIR spectrometry (in contrast to a spectrum recorded with a grating spectrometer, where this assumption is not adequate if photon noise is dominating).
* * *
Line 111: "The assumption of a spectrally smooth phase allows to separate at each spectral position the complex spectrum into two orthogonal components"
"allows to separate" is a construction that is not used in English speaking world. I suggest re-writing as: "The assumption of a spectrally smooth phase allows separation of the complex spectrum into two orthogonal components"

Thanks, we adopted the suggested statement in the revised manuscript.
* * *
Line 112: Please clarify what you mean by "direction".

The revised manuscript now states: the component along the direction **in the complex plane** we expect the spectral signal to be oriented.
* * *
Line 131: "Instead, the ZPD position shifted near one end of the mechanical scan range"
Re-write as "Instead, the ZPD position is shifted to be near one end of the mechanical scan range"

Thanks, done!
* * *
Line 132: I think that "equation (2)" should be "equation (3)".

True, we corrected this in the revised version of the manuscript.
* * *
Lines 151 to 188: This section interrupts the flow of the scientific discussion, by providing a lot of mostly irrelevant information about the various Bruker spectrometers. For example, the fact that they use the "Rock-Solid" design is mentioned twice here, as is the fact that "more than 100 units are sold". This section should be considerably shortened, or moved to an appendix. Perhaps cite the Bruker website/brochure for readers who want this type of information.

TODO: significantly shorten spectrometer descriptions! Add table!
* * *
Lines 191-195: "IZO is managed by the Izaña Atmospheric Research Centre (IARC, https://izana.aemet.es/, last access: 5 August 2024), which belongs to the State Meteorological Agency of Spain (AEMet). Within the IZO's atmospheric research activities, the FTIR programme started in 1999 in the framework of a collaboration between AEMET and KIT [Schneider et al., 2005], contributing to NDACC and TCCON networks since 1999 and 2007, respectively."
I don't find this mostly historical information scientifically relevant. Which organizations manage and fund the IZO site, and how long they have done so, are not relevant to phase correction. Most of this should be moved to the Acknowledgements. The website https://izana.aemet.es contains nothing of relevance to phase correction. Also, the paragraph doesn't mention COCCON, which I thought was the main driver of the new phase correction method.

TODO: significantly shorten spectrometer descriptions! Add table! Mention COCCON
* * *
Line 170: "1mm" ❼     "1 mm". In most places a space is left between the value and the unit, but here not.

Thanks, corrected!
* * *
Line 198: "Schneider etThis al. " ❼     "Schneider et al."

Thanks, corrected!
* * *
Lines 202-215: "Because the phase spectrum across such a region is strongly impacted by the overlapping contributions to the phase emerging from either side of the opaque region, the outcome for the phase at a certain spectral position in the region with reduced transmission will depend on the user-selected resolution for the phase calculation and the chosen apodization function."
I don't dispute this statement – this is what you found. But it doesn't explain why the phase correction operator is so sensitive to the user-selected resolution or apodization. The exact phase value will of course have some dependence on the interferogram points selected (i.e. the phase resolution) and their relative weighting (i.e., apodization). But why does this have a large effect on the phase?
Also, It seems contradictory that points far from ZPD contain some information about the phase, that you wish to retain, but you subsequently fit a low-order polynomial through the phase, which is smoothe, and therefore contains no high-resolution frequencies coming from igram points far from ZPD.

This is an interesting discussion point. In our feeling, the classical Mertz method simply offers less room for optimal design of the phase reconstruction. If we want to enforce a smooth phase spectrum (there is good empirical evidence that indeed it is very smooth in many FTIR spectrometers on the several 100 $cm^{-1}$ scale), we need to restrict the section effectively used from the interferogram to an extremely narrow region centered on the ZPD point. We can vary the weights of the contributing interferogram points in the Fourier transform to some degree by choosing between different apodization functions, but not too much, as we need to wipe out any spectral traces from a boxcar clipping, so we have to use a very effective (strong) apodisation function. When following the Mertz scheme, we cannot independently adjust the interferogram section used for calculating a complex spectrum and the final phase resolution. As a result, the chosen apodization and window width set the interpolative characteristics across opaque subregions. The Mertz method has no insight into the underlying physics, while the analytical phase scheme can be designed to reflect that the complex spectrum is derived from a smooth phase, while all the higher-resolution spectral variations are due to variation of the amplitude of the complex pointer.

It is certainly true that points far away from the centerburst will contribute very little information to the phase spectrum, but, e.g., imagine the situation that the covered spectral range contains several saturated subregions. Then, due to the beat phenomenon, the centerburst is surrounded by sidelobes, which decay only sowly as function of OPD. The phase information contained in these sidelobes cannot be retained when using Mertz and requesting a spectrally smooth phase. In contrast, the analytical phase method can deliver a complex spectrum at spectral higher resolution, and restrict the phase resolution when fitting a phase model to the complex spectrum.
* * *
Lines 227-229: "This proposed method can fail if the phase difference calculated in step 5 is greater than $\pm\pi$. We did not encounter this situation, but it may occur if the phase slope is very steep and can possibly be avoided by appropriate repositioning of the ZPD point when calculating the Fourier Transform.

This is of concern because when automatically processing thousands of interferograms, how would we know when the ZPD point needs to be repositioned?

We agree this is an important task. Before doing the Fourier transform, an algorithm selecting the interferogram point nearest to ZPD needs to be applied on each interferogram. Often, a search for the maximum absolute value of the interferogram amplitude is sufficient. If the centerburst is strongly asymmetric or very wide, using the cross-correlation with a stored reference centerburst will deliver more stable results for the reference point. We assume that all codes in use (as OPUS, the GGG suite, PREPROCESS, …) do some kind of pre-FFT preparation of the interferogram to ensure that a proper reference point is reliably found even after re-initialization of the interferometer.
* * *
Table 1: "logical array indicating availability of valid phase value " ❼ "logical array indicating validity of phase value"

Thanks, we have changed the table entry accordingly.
* * *
Table 1: Why isn't $s(v_i)$ allocated in step 0?

Sorry and thanks for pointing out: this is an oversight. We have added the allocation of this complex array quantity in step 0.
* * *
Table 1: Perhaps give names to the float array containing phase and the logical array containing validity e.g. Psi and LVALID. This would avoid repeating "logical array value of current position" later in the table (twice).

Thanks for the useful suggestion, we have revised the table accordingly.
* * *
Table 1: "Use the value of the cross product between the normalized complex pointers"
What are "normalized complex pointers"? There is no mention of "pointers" anywhere else in the paper.

We intend to refer to a vector in the complex plane and agree we should avoid the designation "pointer". We have changed this into "vectors in the complex plane".

Table 1: At which step is index $j$ updated? Somewhere it is set to $i$, before $i$ is incremented.

We have added an explicit bookkeeping of the index values i and j in the description of the algorithm (table).
* * *
The equation $\Delta\phi_{(j\rightarrow i)}$ = asin{ } in Table 2 finds the phase difference between the complex vectors $S_{vi}$ and $S_{vj}$ by using their cross product. It is not obvious why this is the case. To me, it would be simpler to directly subtract the phases at $v_i$ and $v_j$ as follows:

Let $S_{vi}$=[a+ib] , $S_{vj}$=[c+id], where i=Sqrt(-1). The phase of $S_{vi}$ is atan[b/a] and that of $S_{vj}$ is atan[d/c].

$\Delta\phi_{(j\rightarrow i)} = \phi_i - \phi_j$ = atan[b/a] - atan[d/c] = atan[(ad-bc)/(ac+bd)]

To me this approach seems more intuitive than using cross-products. So please explain the advantage using a vector product, rather than simply subtracting the two phases.

The very substantial advantage of using the cross product is that this formulation only senses phase differences between the involved vectors (which have similar phase orientation), not to absolute phase orientations of the vectors involved. The use of atan (or even atan2) functions is very dangerous and will occasionally fail (which would require additional tests or case discriminations), as the two vectors might reside on different branches of the inverse function.
* * *
Table 1: Where does the "phase unwrapping" occur? This is mentioned earlier in the paper including the abstract. But then nothing more.

We discuss this point on the first page of this reply (and with the anonymous reviewer as well). In the revised version of the manuscript, we now make transparent that we understand the proposed algorithm to be a two-step procedure, consisting of (1) phase unwrapping, and then (2) fitting the parameters of an analytical phase model to the unwrapped phase spectrum.
* * *
Line 226: Here Table 1 is mentioned, but there is no description in the main body of text. So the table itself and its caption need to be more self-explanatory.

We have extended the table caption: step-by-step procedure **for the phase unwrapping algorithm**, which develops the raw phase used as input for the model fit.
* * *
Line 234: "The second step…". At first, I thought that this referred to Table 1 (Step #). But that doesn't make sense. So perhaps change this sentence to "After completing the raw phase vector over the full bandwidth, as described in Table 1, we next fit the parameters of the phase model to the raw phase values."

Sorry, the lack of clarity results from not presenting in detail the two-step character of the approach. We hope that this problem is removed in the revised manuscript.
* * *
Line 247: Need to clarify whether P*model* is the retrieved model parameters (e.g., polynomial coeffs) or the reconstructed phase values. Line 247 states " P*model* is the set of model parameters". But line 250 states "after receiving the set of model parameters, P*model* can be calculated at all spectral positions", which suggests that P*model* is the reconstructed phase values themselves.

Thanks for pointing out this error: you are right in that P_*model* are the retrieved model parameters. In the second sentence, it should be phi_*model*, referring to the phase resulting from the analytical model after fitting of the model parameters.
* * *
Line 256: "phase calculation uses a resolution of about 10 $cm-1$, which is supported by all spectrometers we included in the study (sufficient number of points on the short side of the interferogram)." It would be helpful to some readers to also express this as interferogram points. Assuming the usual two samples per reference laser wavelength, I think that 10 cm$_{-1}$ corresponds to a MOPD of 0.9/10=0.09 cm which is 0.0/0.3154E-04=2844 interferogram points on each side of ZPD. If correct, perhaps state this number.

Thanks, we have added the information that in our current implementation a range of 3000 points on either side of ZPD is used: The raw phase calculation uses 3000 interferogram points on either side of ZPD, equivalent to a resolution of about 10 cm-1, which is supported by all spectrometers we included in the study (sufficient number of points on the short side of the interferogram)

Figure 1: Why does the right-hand panel have a slightly narrower wavenumber range that the left-hand panel, or figs. 2, 3, 4.

Sorry, this was unintended, the plotting software created some differences between the displays. In the revised version of the manuscript, we have aligned the coverage of all plots (spectral range now consistently 4000 - 12000, increment 2000 cm$^{-1}$).
* * *
Line 269: "model (analytical, ana)". What is "ana"? I puzzled over this for a while before deciding that you are defining a new abbreviation. But why not simply label the y-axes of the figs "analytic - raw" – there's plenty of room. The "ana" abbreviation isn't used anywhere else.

Correct, thanks for pointing out. We now refer to "analytical" in the revised figures and drop the abbreviation "ana".
* * *
Line 269: Is the "fitted model (red)" the same thing as "model (analytical, ana)"? If so, use the same terminology. If they are different, explain the distinction.

Ok, we now consistently refer to "raw phase (black) and fitted analytical phase (red)" in the figure captions.
* * *
Line 283: "The curvature of the phase is somewhat stronger than in case of the IRCube." ❼
"The curvature of the phase is somewhat stronger than in **the** case of the IRCube."

Thanks, corrected.
* * *
Line 308: "Figure 5 shows the effect of using either the Mertz or the analytical phase " ❼ "Figure 5 shows the effect of using either the **classical** Mertz or the analytical phase…"

Ok, changed as suggested.
* * *
Line 316: "This reminds of the fact that…" ❼ "This is a reminder of the fact that"

Ok, changed as suggested.
* * *
Line 320: "…phase reconstruction is of higher importance for single-sided interferograms (all the spectrometers investigated here apart from the EM27/SUN) than for the EM27/SUN, which essentially is insensitive to phase errors in reasonable limits" This is a bit clumsy. I suggest re-writing as: "…phase reconstruction is much easier and accurate for double-sided interferograms (i.e., the EM27/SUN) than for single-sided interferograms (the other spectrometers discussed here)"

Here, we would prefer to maintain the original statement. The main concern we primarily intend to point out here is not that phase reconstruction is easier and more accurate for double-sided interferograms, but that the propagation of a certain amount of phase error into spectral domain is much less detrimental in case of double-sided interferograms. If the interferogram is double sided, the introduction of a phase error is (locally) simply a scaling of the spectrum, while for a single-sided

interferogram any spectral features (absorption lines) are strongly distorted (due to the fact that the cancellation of sine contributions originating from – OPD and + OPD cannot cancel out any more). The fact, that the spectral differences are ways smaller for the EM27/SUN than for the IRcube is primarily due to this mechanism, while the residual phase errors (analytical phase versus truth) are not expected to be very different.
* * *
Line 323-326: Split into two sentences.

Ok, done!
* * *
Line 326: Needs a reference for why atmospheric xCO2 measurements need an accuracy of 0.05 ppm.

We unfortunately do not have a citable reference, but in practice find ourselves regularly confronted with requirements on this level. The planned CO2M GHG sensor aims at an accuracy level of 0.5 ppm for $XCO_2$ (Rusli et al.: "Anthropogenic $CO_2$ monitoring satellite mission: the need for multi-angle polarimetric observations", AMT, 2021), and EUMETSAT expects the ground-based reference networks to be significantly more accurate ("by about one order of magnitude"). We agree this requirement might be somewhat excessive and therefore arguable, but the expectations of our stakeholder are obviously high.

A more comprehensible line of argument, which produces a comparable 0.05 ppm requirement is related to the quantification of $CO_2$ sources. A medium-large city as, e.g., Munich, only occasionally generates differential column signals beyond 1 ppm, the majority of observations resides on the 0.5 ppm for typical wind situations. If we desire that the uncertainty of reconstructed source strength is dominated by the uncertainty of transport (wind fields and vertical mixing) and if we assume a reasonable 20% uncertainty for this contribution, this requires an uncertainty budget of the FTIR measurements on the 10% level, so 0.05 ppm.

We added the following explanation in the manuscript:

For example, Rißmann et al., 2022, state that the XCO2 gradients across the medium-sized city Munich typically are well below the 1 ppm level. Let us assume 0.5 ppm as a typical signal amplitude and the uncertainty on the source strength estimate due to imperfect description of transport to be on the 20% level. In order to avoid a significant uncertainty contribution from the FTIR observation, we need an accuracy level of $10\% \cdot 0.5\ ppm = 0.05\ ppm$.
* * *
Line 328: Is $2 \cdot 10^{-5}$ a fractional change? So this is a $425ppm*2\cdot10^{-5} = 0.0085$ ppm change in xCO2. This is very small. So for the EM-27 or the IR cube, the classical Mertz phase correction is perfectly adequate.

True, the reconstruction of the phase spectrum is not a demanding task for the $CO_2$-band used. As emphasized by you before, nearly opaque spectral regions are the difficult targets and should be used to demonstrate the effects more clearly. We have extended the discussion in section 5 accordingly. The TCCON is in the process of extending its spectral coverage into the mid infrared spectral range. We expect that spectral information from stronger mid-infrared bands of $CO_2$ and $CH_4$ will be combined with the standard bands in the near infrared. In order to illustrate the associated challenge for phase reconstruction in such near-opaque spectral regions, we now include an investigation of the 8730 – 8850 cm$^{-1}$ window and show the resulting effects on retrieved XH2O.
* * *
Line 333: How much difference between the two phase correction methods for the Vertex and IFS125? These instruments seem to have been dropped from the discussion.

We included a wider set of FTIR spectrometers just to show which level of variability is found for the characteristics of the phase spectrum between different types of spectrometers. The most unusual behavior is that of the EM27/SUN, which has a nearly linear phase, all other spectrometers show similar degree of curvature in their phase spectra (in case of the Vertex, the analytical phase reveals some superimposed undesired oscillatory phase behavior in addition). Our study does not aim at a systematic investigation of phase error propagation into XGAS, but we now have included an example using the IRCube to show that the differences between Mertz phase and the proposed scheme are relevant in saturated bands.